# Point Cloud Deep Learning Network Based on Balanced Sampling and Hybrid Pooling

**DOI:** 10.3390/s23020981

**Published:** 2023-01-14

**Authors:** Chunyuan Deng, Zhenyun Peng, Zhencheng Chen, Ruixing Chen

**Affiliations:** School of Electronic Engineering and Automation, Guilin University of Electronic Technology, Guilin 541004, China

**Keywords:** S3DIS, weighted sampling method, self-attention model, hybrid pooling

## Abstract

The automatic semantic segmentation of point cloud data is important for applications in the fields of machine vision, virtual reality, and smart cities. The processing capability of the point cloud segmentation method with PointNet++ as the baseline needs to be improved for extremely imbalanced point cloud scenes. To address this problem, in this study, we designed a weighted sampling method based on farthest point sampling (FPS), which adjusts the sampling weight value according to the loss value of the model to equalize the sampling process. We also introduced the relational learning of the neighborhood space of the sampling center point in the feature encoding process, where the feature importance is distinguished by using a self-attention model. Finally, the global–local features were aggregated and transmitted using the hybrid pooling method. The experimental results of the six-fold crossover experiment showed that on the S3DIS semantic segmentation dataset, the proposed network achieved 9.5% and 11.6% improvement in overall point-wise accuracy (OA) and mean of class-wise intersection over union (MIoU), respectively, compared with the baseline. On the Vaihingen dataset, the proposed network achieved 4.2% and 3.9% improvement in OA and MIoU, respectively, compared with the baseline. Compared with the segmentation results of other network models on public datasets, our algorithm achieves a good balance between OA and MIoU.

## 1. Introduction

Due to the disorder, nonuniformity, and irregularity of point cloud data, semantic segmentation of point cloud data is a challenging task [1] and is a current research focus. Although deep learning methods have achieved considerable success on semantic segmentation tasks, deep learning methods based on convolutional neural network (CNN) [2] can only perform feature extraction on 2D images; they cannot directly process unstructured raw point clouds. Therefore, experts and scholars have divided point cloud semantic segmentation tasks into two categories: indirect and direct point cloud processing.

In indirect point cloud processing methods, point cloud data are first converted into multiview [3] or regularized voxel data [4], and then neural networks are used to learn 2D image features or voxel-level 3D features to achieve semantic segmentation. Such methods have many limitations and strict requirements on the uniformity of point cloud density. Detailed information is lost from the sparse areas of point clouds in the process of view and voxel conversion, whereas the dense areas of the point cloud experience label information stacking, and the multiview and voxel conversion methods cannot completely express the dense areas, this results in low-accuracy semantic segmentation. Additionally, when multiview and voxelized point cloud semantic segmentation methods are used with massive point cloud data and in complex information scenarios, the graphics processing unit (GPU) occupancy rate and computational cost exponentially increase.

In direct point cloud processing methods, a single point cloud is used as the target to extract features to achieve the feature learning and semantic segmentation of unstructured point cloud data. Its pioneering representative is PointNet [1], but this network only learns the features of local area sampling center points, ignoring the local fine-grained features composed of its neighborhood points. Therefore, Qi et al. [5] proposed PointNet++ on the basis of PointNet. The network uses FPS to sample in the K neighborhood and groups the sampling points for feature downsampling of different dimensions, thereby constructing a hierarchical network to extract local features. However, PointNet++ still extracts features from independent points and ignores the deep feature relationship between the sampling center point and its neighbors.

To solve the above problems and especially to improve the network’s ability to process unbalanced data, the network model proposed in this paper provides three contributions:On the basis of PointNet++, the network uses a weight-based balanced sampling module to replace the original data sampling module, which balances the network’s ability to extract various sample data.We designed a self-conv (SC) module to be integrated into the downsampling layer of the original network to further improve the learning ability of the network for local fine-grained features, which mainly includes the following three aspects: (a) Feature learning of sampling center points and neighbor points: Aiming at the shortcomings of the current point cloud semantic segmentation network and focusing on describing the spatial relationship with the Euclidean distance and direction vector in the downsampling stage, the positional relationship between points is used as the intensity value and eigenvalue fusion, so as to enrich the spatial features of the sampling center point. (b) The mutual structural relationship learning between neighborhood points and neighborhood points: At present, most of the existing networks ignore the feature information of the neighborhood space. To solve this problem, we propose a vector calculation between the neighborhood point and the minimum neighborhood point to further enrich the features obtained in the downsampling stage. (c) Local feature enhancement under the attention model: The features of each sampling center point are different and incomplete. To solve this problem, we used an attention model to “score” different features and synthesize them according to their importance to ensure the complete expression of the spatial saliency structure.The hybrid pooling method is used to predict the segmentation and loss value calculation of the feature model output by the spatial pyramid pooling module and the max pooling module. Comparing their loss values, the weight matrix corresponding to the minimum loss value is taken and sent to the next iterative training epoch. The results of our experiments showed that this network design improves the accuracy of semantic segmentation and is more effective than other direct point cloud semantic segmentation algorithms.

## 2. Related Studies

### 2.1. Multiview-Based Point Cloud Segmentation Methods

Huang et al. [6] developed a method that learns local descriptors from multiview correspondences to enhance network generalization. Feng et al. [7] further proposed a GVCNN framework to group different view visual descriptors to effectively use feature multiview relationships. Boulch et al. [8] back-projected the projected view pixel labeling results onto the original point cloud to achieve multiview feature aggregation and joint processing. Although multiview projection methods endow the point cloud with structural features and are convenient for network implementation, they destroy the inherent geometric relationship of point cloud 3D data and lose key spatial information, thus affecting the accuracy of semantic segmentation.

### 2.2. Voxel-Based Point Cloud Segmentation Methods

Point cloud voxelization processing, in terms of volume discretization, better maintains the inherent geometric relationship of its data. The 3D convolutional neural network represented by VoxNet fully learns the 3D spatial context information for segmentation purposes, but the selection of the size of the voxel block affects the performance of the network. If the voxel block is too large, details are lost; if it is too small, the computational burden is increased.

### 2.3. Direct Point Cloud Processing Methods Based on PointNet

One method, 3DRCNN [9], uses a point-wise pyramid pooling module to capture multiscale domain local features. A bidirectional hierarchical recurrent neural network (RNN) is then used to explore long-range spatial dependencies. Li et al. [10] proposed PointCNN, which uses a learnable matrix X to weight and permutate 3D point cloud data, which not only preserves the spatial location information of the point cloud but also obtains spatial local features. NormNet [11] proposed a multiscale K-nearest neighbor (K-NN) convolution module based on PointNet. The network successfully fits global, semantic, and local features of multiscale K-NN convolution modules. Local feature enhancement and network noise suppression were realized. Zhao et al. [12] designed an adaptive feature aggregation module based on PointNet++. In the downsampling stage, a relationship calculation module between the center point and the neighborhood point is added, and the calculated relationship feature is compared with the original sampling center. The point features are aggregated to strengthen the network’s learning ability for point cloud features. HDA-PointNet++ [13] and DPE [14] improve the network’s learning ability of large-scale category features and deep point features, respectively, through hierarchical data enhancement. The overall segmentation performance of these methods on public datasets was improved compared with that of previous methods, but they could not effectively distinguish semantic classes with extremely similar overall geometric shapes and slightly different local detail structures, showing different degrees of undersegmentation problems.

### 2.4. Direct Point Cloud Processing Method Based on Graph Convolution

Wang et al. [15] applied the graph convolutional neural network (GCNN) to point cloud processing for the first time and constructed a dynamic graph convolutional neural network (DGCNN) using an edge convolution operation to extract center points, as well as features and edges of center and neighboring points. Landrieu et al. [16] used a super point graph to capture the contextual relationship between the local spaces of 3D point clouds, and then fed the extracted information into a graph convolutional neural network for feature learning to achieve point cloud semantic segmentation. GACNN [17] is a graph convolutional neural network based on a global–local attention mechanism. First, the network obtains the spatial relationship between all points through the global attention module, then dynamically learns the convolution weights according to the spatial positions of local adjacent points, and uses the density of each local area to perform weighted updates on the convolution weights. Finally, the multiscale features of the point cloud are captured to achieve on-board point cloud segmentation. GraNet [18] is a global relation-aware attention network. A local spatial difference attention convolution module and a global relation-aware attention module are embedded in the network’s multiscale feature learning framework to enhance the deep features of long-range dependencies and achieve urban point cloud segmentation. In general, the performance of point cloud semantic segmentation networks based on GCNN is excellent, but the number of GCNN nodes is related to the number of point cloud points, and the network structure is relatively fixed, so large-scale point cloud processing is a problem.

In this study, we used PointNet++ [5] as the semantic segmentation framework; its frame structure is shown in Figure 1. Compared with other point series networks, this network is characterized by a clear structure, point-to-point relationship learning framework, and short running period. The FPS algorithm of the network can extract the outer contour structure of the target object as much as possible, ensuring that the downsampling does not lose too much spatial structure information. The advantage of FPS over random sampling is that it can cover all points in the space to the greatest extent possible. When the majority class samples with a large number of point clouds must be downsampled, the FPS algorithm not only reduces the number but also retains the spatial feature information of the majority class samples. PointNet++ also uses K-NN grouping and max pooling layers to capture the neighborhood information of the sampled center points. However, the max pooling layer only relies on a fixed window stride and cannot fully use the global information. To more effectively learn the fine-grained local features of the point cloud, researchers [19] used the attention model to adaptively filter the local features and used PointConv to project the relative positions of the two points to the weight of the convolution to obtain local fine-grained information. Based on the above literature, we designed the SC module to analyze and learn the relationship between the sampling center point and its K neighborhood points, and we introduced an attention model into the feature aggregation module to improve the network’s perception and learning of fine-grained features at different scales. Additionally, we introduced a spatial pyramid pooling framework [20] to pool the features of each dimension obtained by downsampling under different window sizes, which is parallel to the max pooling method under the fixed window size of the original network. Figure 2 demonstrates the spatial pyramid pooling structure.

The spatial pyramid pooling (SPP) structure in Figure 2 can accept feature maps in different dimensions, and it pools the feature maps of each dimension to output a unified dimension feature model for predicting segmentation and classification. This structure modifies the connection between layers on the basis of the traditional image pyramid algorithm and fuses local and global features without increasing the amount of calculation of the original model. The results of experiments have shown that the fusion of SPP with residual network [2], faster R-CNN detector [21], YOLO V3 [22], or RPN [23] can improve the performance of the original network in terms of object detection, image classification, and object segmentation. Based on the above literature, we integrated the spatial pyramid pooling structure in the pooling process, and we interactively fused the features extracted by the network across spaces and scales to obtain rich spatial context information.

## 3. Materials and Methods

In this paper, we propose a point cloud deep learning network based on balanced sampling and discrepancy pooling, which consists of three modules: (1) Weight-based balanced sampling module. This module samples the training set, generates the corresponding sampling initial weights, then performs feature learning and model output on the sampled data set, and calculates and iterates the loss function value of each training batch, finally obtaining a relatively balanced sampling data set. (2) SC feature extraction module: This module learns and aggregates the spatial position relationship between the sampling center point and its neighborhood points obtained by the downsampling layer, so that the features contain rich spatial structure information, and self-attention scoring and aggregation of the features are encoded by each sampling point. (3) Hybrid pooling module: First, a spatial pyramid pooling layer is used to send the features of each dimension obtained by downsampling into pooling windows of different sizes for pooling. Second, the results are then evaluated in parallel with the output model of the max pooling layer, and the optimal network parameters are returned to achieve hybrid pooling.

### 3.1. S3DIS and Vaihingen Dataset

The proposed algorithm uses two large 3D point cloud segmentation datasets for validation: In-Cloud Stanford Large-Scale 3D Indoor Space (S3DIS) [24] and the International Society for Photogrammetry and Remote Sensing (ISPRS) Vaihingen 3D semantic segmentation competition [25] datasets.

The S3DIS, published by Stanford University, includes 6 areas of 3 different buildings, which are divided into 271 independent rooms. Each point in the scene corresponds to a fixed label, which belongs to 13 categories (ceiling, floor, wall, door, etc.). The S3DIS dataset has a total of 6 regions, of which the total number of points in regions 1–5 is 26,489,056. The proportion of each category and the total extracted data are shown in Table 1.

Table 1 shows that S3DIS is an unbalanced data set with ceilings, floors, and walls accounting for the majority samples and the other 10 classes being the minority. In comparison, the Vaihingen dataset was manually segmented into 9 types of objects, including façades, impervious surfaces, and shrubs. The data used for training are spliced from five subareas, and the test set includes commercial housing and villa areas. The detailed information of the various objects in the training data set is shown in Table 2.

Table 2 shows that the sample data categories are not balanced, with 87.785% of the sample being trees, roofs, low vegetation, and impervious surfaces. However, due to the limited geometry and distribution features of a few categories such as power lines, shrubs, and hedges, their total number of points is far less than for other types of data. Additionally, the linear distribution of the features further leads to fewer minority class points in the sub-block. Usually, PointNet++ series networks aim at overall optimality and tend to sacrifice the segmentation accuracy of minor targets when dealing with imbalanced samples. Therefore, we needed to adjust for extremely imbalanced data to improve the balance of each category.

### 3.2. Weight-Based Balanced Sampling Module

The point cloud data converted for the real world often do not obey a uniform and balanced distribution. The traditional learning method aims at reducing the OA and treats all samples equally, which results in the network having high segmentation accuracy in the majority class and low segmentation accuracy in the minority class. The OA of the final model is good, but the minority class feature information is mostly ignored. As such, the model does not meet the requirements. From the perspective of data sampling, experts and scholars have created solutions to such problems that can be classified as undersampling, oversampling, and mixed sampling.

The undersampling methods [26,27,28] conditionally extract some samples from the majority class and the popular class samples. Their limitations are that the final training set loses data, and the model only learns part of the overall features. Due to discarding a large amount of data, these methods experience problems such as overfitting, missing feature information for most class samples, and overall classification accuracy decline. The oversampling methods [29,30,31,32] copy the niche samples multiple times, and a point repeatedly appears in the high-dimensional space, which leads to strong randomness in the probability of the point being correctly or incorrectly classified. The repeated and interpolated samples also cause problems such as overfitting, noise interference, and overgeneralization during the training process. The basic idea of mixed sampling is to expand the number of samples in the minority class and weaken the total amount of samples in the majority class so as to balance the data. However, the challenge with this method is setting the balance between weakening and expansion. Based on the above literature summary and analysis, we developed a weighted sampling method that can adapt to the data and network in this study. The algorithm flow is as follows:

Input: All training samples, maximum number of iterations T, loss function L, log-likelihood cost function.

1. Extract all minority class samples and count the total number of point clouds as the FPS sampling algorithm parameters, use FPS to downsample the majority class samples. The sampled majority class samples and minority class samples form a training data set I = [(*x*_1_, *y*_1_), (*x*_2_, *y*_2_), …, (*x_i_*, *y_i_*)]; set the initial value of the weight *ω*(*i*) of various samples in I.

2. For t = 1, 2, …, T:

a. Import the training data set I and weight *ω*(*i*) into the network for model training, evaluate the loss value of the training model, and store the training model, loss value *l*(*t*), and the corresponding weight *ω*(*i*) in Set A.

b. Extract the label set *h*(*t*) of various samples in the training model, calculate *h*(*t*), and judge the ith sample as TP(*i*) of the actual category of the sample. Judge it as FN(*i*) of the opposite category to the actual category of the sample.

c. Calculate the error *e*(*i*) = ∑*ω*(*i*) × [1 − TP(*i*) + FN(*i*)] of each category of samples, and calculate the total error *e* = ∑*e*(*i*). If there is *e* = 0, then stop the loop and save the current sampling weight *ω*(*i*) as the final weight value; otherwise, calculate *α*(*t*) = *e*/(1 − *e*) and proceed to the next step.

d. Update the weight, *ω*(*i*) = *ω*(*i*) × *α*(*t*)^(0.5 × [1 + TP(*i*) − FN(*i*)]); return the new weight value.

The method solves the problem of data imbalance by adjusting the sampling weights. When initializing the weights, larger weights are assigned to minority class samples and assign smaller weights to majority class samples. The initial weight is evaluated and adjusted through the loss function value fed back by the training model, and the optimal sampling weight is iteratively obtained. Finally, the training samples of the network tend to be balanced.

### 3.3. Network Structure Design

When the deep learning algorithm with PointNet as the baseline processes large-scale 3D point cloud data, the large scene point cloud data are divided into different sub-blocks. Then, the original point cloud in the domain is sampled and sent to the network in batches for feature extraction [1]. In this study, we used weight-based balanced sampling to maintain the balance between the point cloud data structure in the sub-block and the overall sampling sample as much as possible. However, differences exist in the density of the point clouds in different regions, and the point clouds tend to be sparse after sampling, so that the adjacency relationship of point clouds is somewhat destroyed, which affects the full expression of the local geometric structure features of the target and the learning ability of the network for this feature. To solve the above problems, researchers [33] proposed a method of combining self-attention with deep learning, which can learn its own representation through only a single sequence in natural language processing tasks, which considerably increases the network’s ability to learn the feature. Additionally, max pooling is suitable for extracting local features such as edges, lines, and textures in feature maps, whereas spatial pyramid pooling is suitable for aggregating global features of multiscale feature maps. Fitting global and local features helps to improve the performance of a network, as well as generalization and segmentation accuracy. Authors [34] proposed a differential pooling method based on probability distribution that can effectively fit different pooling modules, so that the network has better generalization and robustness in classification tasks.

In the proposed method, the feature extraction network is based on PointNet++, and we introduced the attention model and hybrid pooling while focusing on two factors. The first was to encode features for the spatial topological relationship and use the self-attention model to score and aggregate the features of different sampling points. The second was that existing networks generally use max pooling for feature dimensionality reduction and transmission, and generally lack the pooling method that combines global and local features, which may improve the network model’s ability to express global features such as color, texture, and overall shape. The overall structure of our network is shown in Figure 3.

#### 3.3.1. SC Feature Learning Module

In Figure 3, *N* is the number of central sampling points, *D* is the feature dimension of each point, and *D* = 3 represents the input three-dimensional point cloud data. The input point cloud with dimension *N* × *D* is used for center–neighborhood and neighborhood–neighborhood feature learning by the SC module, as shown in Figure 4. In the SC module, the input point cloud is first sampled by FPS to obtain the target center point *N_i_* × 3, then the KNN neighborhood point sampling *N_i_* × *K* × 3 is performed on the center point, and then the two pieces of point cloud data are input into the combine module for center point and neighboring point sampling. Domain point relationship learning and feature extraction are performed to obtain the feature information with dimensions *N* × *K* × *D*; then, the data are put through the MLP and maximum pooling layers to obtain the feature information of dimension *N* × *D*. Finally, the point cloud is sampled by FPS and by KNN, and the feature information of neighborhood relationship *N* × *K* × (3 + *D*) dimension features is obtained by splicing with the max pooling feature information. As shown in Figure 3, the input data pass through the first set of SC modules to output feature information with *N*/4 × 64 dimensions; after passing through the second, third, and fourth sets of SC modules, *N*/16 × 128-, *N*/64 × 256-, and *N*/256 × 512-dimensional feature information is obtained, respectively. The feature information is then aggregated through the spatial pyramid pooling module, so that it contains both multiscale local regional features and global features. Finally, the pooled features of each layer are spliced and input into the fully connected layer to obtain the label classification of each point to achieve semantic segmentation. The output is the score of m categories to which each point in the point cloud belongs.

The input in Figure 4 is *N* point clouds containing *x_i_*, *y_i_*, and *z_i_* spatial coordinate information. *N_i_* sample center points are obtained under FPS, and $N_{i}^{k}$ neighborhood points are obtained under KNN. Then, the encoding method of the spatial feature information of each sampling center point *N_i_* is as follows:
(1)$$\beta_{i}^{k}=MLP(N_{i}⊕f(F_{i},\;F)⊕N_{i}^{k}⊕f(N_{i}^{k}))$$

The sum in the formula is the feature enhancement module of the sampling center point and neighborhood point and the feature enhancement module of the neighborhood point and domain point. The main purpose here is to compare and analyze the characteristics of the sampling points and sampling neighborhood points, and to aggregate and extract the feature relationship between them. Given the input feature set *F* = [*F_i_*, *i* ∈ (1, …, *G*)], we use the following formula to enhance each feature in the set:
(2)$$f(F_{i},\;F)=\sum_{j=1}^{G}f_{gx}(F_{i},\;F_{j})\cdot f_{db}(F_{i},\;F_{j}),\;f_{gx}(F_{i},\;F_{j})=F_{i}-F_{j},\;f_{db}(F_{i},\;F_{j})=MLP(g(F_{i},\;F_{j}))$$
where *g*(·) is the one-dimensional convolution operation, which is used for feature fitting, and is the relationship function between them. The relationship strength of the two is multiplied with the feature information to perform the feature enhancement operation. It mainly calculates the spatial displacement vector between the neighborhood point and the minimum neighborhood point, and we use the following formula to express the spatial topological relationship between the neighborhood points:
(3)$$f(N_{i}^{k})=N_{i}^{k}-\min(N_{i}^{k})\;|k∊[0,\;K]$$

With Formula (1), the network not only obtains the feature information of the sampling center–neighborhood point, but also fully absorbs the neighbor features between the neighbor points. We introduce a self-attention model to the proposed network to enhance the learning ability of the network model to obtain the saliency structure (feature) of the neighborhood from the sparse sampling point neighborhood. The module is shown in Figure 5.

As shown in Figure 5, the network uses the attention module to calculate the attention score of the feature $\beta_{i}^{k}$ of the sampling center point:
(4)$$A_{i}^{k}=MLP(softmax(\beta_{i}^{k},\;W))$$
where *W* is the shared learning weight of the multilayer perceptron. After obtaining the attention score of each sampling center point feature, both are aggregated to obtain the attention feature containing contextual information. The formula is as follows:
(5)$$AT(N_{i})=\sum_{k=1}^{K}A_{i}^{k}\times c(\beta_{i}^{k})$$
where *c*(·) represents the 1 *×* 1 convolution, and the multiplication of its corresponding attention score is the feature vector $AT\left(N_{i}\right)$ output by the SC feature learning module during the downsampling process.

#### 3.3.2. Hybrid Pooling

In the PointNet++ structure, the upsampling stage performs multiple sampling, grouping, convolution, and pooling of features at different scales, and finally unifies the dimensions of the downsampled features of each layer and splices them together to obtain a segmentation model. Although this method obtains the local fine-grained features of each point, multiple sampling and pooling reduce the integrity of global features to a certain extent. Therefore, we designed a new hybrid pooling layer based on spatial pyramid pooling and max pooling. The spatial pyramid pooling layer designs the pooling window of its corresponding step size for the features at each scale; the formula is as follows:
(6)$$Pool(f_{i})=MLP(conv(Pyramid((AT(N_{i},\;B_{i})\;|i∊(1,\;G))))$$
where $B_{i}$ is the window step size corresponding to feature *AT*; *Pyramid* is a spatial pyramid pooling operation, which performs max pooling of features of different dimensions and superimposes them by cancat splicing; *Conv* is a convolution operation, and its role is to merge the multiscale features obtained by spatial pyramid pooling. Therefore, the output of spatial pyramid pooling contains both multiscale local region features and global features. However, complex scene data usually have issues such as feature aliasing, object occlusion, and structure mosaic; the max pooling layer has good applicability to local scene data with such problems. Therefore, as shown in Figure 3, the features under the self-attention model are separately sent to the max pooling and spatial pyramid pooling for feature aggregation output, and the models they output are predicted and segmented on the test data set to obtain two different loss values, *L_i_* and *L_j_*; the weight matrix of the smallest one is taken and sent to the next training epoch.

## 4. Experiment and Analysis

### 4.1. Experimental Environment and Evaluation Index

We conducted our experiments based on the Tensorflow 1.13.1 platform, which depended on a TiTAN XP GPU. During the training, we used the Adam optimizer with default parameters. We set the initial learning rate as 0.0002 and the batch size to 128. We used the exponential decay method to control the changes in the learning rate. We set the momentum, decay period, and decay rate to 0.5, 10,000, and 0.92, respectively. For all datasets, we trained for 100 epochs. For the evaluation metrics, we used the mean of class-wise intersection over union (MIoU), overall point-wise accuracy (OA), and balanced *F* score (*F*_1_ score). The calculation formulas are as follows:
(7)$$\mathrm{MIoU}=(1/k)\cdot \sum_{i=0}^{k}p_{ii}/(\sum_{j=0}^{k}p_{ij}+\sum_{j=0}^{k}p_{ji}-p_{ii}),\;\mathrm{OA}=n/N,\;F_{1}=2p_{ii}/(\sum_{j=0}^{k}p_{ij}+p_{ji})$$
where *k* represents the number of categories of the point cloud in the data set (including empty categories), *p_ii_* represents the point in the point cloud data with category label *i* and the actual predicted category *i*, *p_ij_* indicates the category label is *j*, the actual predicted category is *i*, *p_ji_* represents the point whose category label is *i* and the actual predicted category is *j*, *n* represents the number of all correctly predicted points, and *N* represents all points in the point cloud. The larger the MIoU and the larger the OA, the better the segmentation effect and MIoU <= OA. The *F*_1_ score is defined as the harmonic average of precision and recall, and its value ranges from 0 to 1. The closer the *F*_1_ score is to 1, the better the segmentation effect on class *i*.

### 4.2. S3DIS Dataset Experiment

#### 4.2.1. Crossover Trial

Our aim with the experiments described in this section was to verify the effectiveness of the sampling, feature encoding, and pooling modules described above. Taking regions 1–5 of the dataset as training samples, we divided the room into 1 × 1 m subareas without considering the elevation distribution during training, and we randomly selected 4096 points in each subarea to generate the training data. Using PointNet++ as the baseline, we designed the weight-based balanced sampling (ES), SC feature encoding (ATT) under the self-attention mechanism, pyramid pooling (PYA), hybrid pooling (HYB), and full-module combination (ALL) ablation experiments (Table 3) to analyze the impact of each module on the segmentation accuracy of Region 6. Table 4 shows the segmentation results under the corresponding module combinations; Table 5 shows the segmentation effect of the different components("√" indicates that the component is enabled).

Table 4 shows that compared with the baseline’s MIoU (approximately 70.2%), when only the ES module is considered in the model, the segmentation accuracy of other categories except for floor, beam, door, and window was improved to a certain extent. The reason for this is that the areas bordering the wall and door have similar structural features. When the wall sampling points are diluted, the feature information extracted by the wall in the baseline network is missing, resulting in a small part of the wall points being classified as door points. The ratio of points to wall and door is 4.9:1, so the IOU of the door category dropped. The ceiling and beam produced similar results, with the difference being that the ceiling is only adjacent to the wall and the structural features of the two are quite different, so the IOU of the ceiling did not drop but increased. We found that ES could effectively improve the ability to learn the features of the minority class samples while ensuring the stability of the classification accuracy of the majority class sample. For the combination of the spatial pyramid pooling module and the baseline network, the MIoU was 72.1%, and the segmentation accuracy of all categories except from beam and window was improved. In particular, the segmentation accuracy on three types of objects, column, bookcase, and clutter, improved by 5.3%, 6.3%, and 5.9%, respectively, compared with the baseline. This fully demonstrates that the ability of the spatial pyramid pooling module to extract local spatial relationships and global features is superior to that of the baseline’s max pooling module. When the ATT module is combined with the baseline, the MIoU improved by 3.1%. The extraction effect for bookcase, sofa, and clutter was particularly notable. For these three types of objects with high feature overlap, the ATT module pays attention not only to the features between points in the domain, but also to learning the mutual relationship and spatial influence between them. Finally, ATT focuses and fuses these features to highlight the distinction between objects of various categories, thereby improving the OA. The combination of baseline with the HYB module obtained 75.8% MIoU, and the optimal IoU was obtained for two types of objects: wall and door. Combined with the HYB module, we found that after obtaining the minimum loss value of the two pooling methods, the weight matrix corresponding to the minimum loss value was returned, thereby realizing the dual expression of global shape features and spatially local fine-grained features. The data showed that the integration ability of gold pooling for global features and the aggregation ability of the max pooling module for local features both have their own advantages in semantic segmentation, and they form complementary advantages under the differential pooling module. Furthermore, the MIoU that was obtained by integrating each module combination into the baseline was increased to 80.0%. This indicated that the ES, ATT, and HYB modules could be effectively combined.

The first and fifth rows in Table 5 show that the ES module could effectively balance the situation of various types of data being sampled, which considerably increased the possibility of a few types of samples being learned, thereby improving the MIoU. When the ES module was omitted, most samples’ classes were more captured and learned, and their segmentation accuracy was improved, thus slightly improving OA. The second and fourth rows of Table 5 show that when we removed the PYA module, the network lost global information when it transmitted features only through maximum pooling. Introducing PYA but not using the HYB module to optimize and filter the two pooled results led to redundant output feature information, thus affecting the key feature representation. According to Table 4 and Table 5, the ES module extracted the overall training samples as balanced as possible, the ATT module covered and learned the spatial structure of the sampling center point in a carpet-like manner, the PYA module transmitted the learned features in a hierarchical manner, and the HYB module ensured the effective transmission of all categories of feature information from PYA and the maximum pool output to a certain extent, thereby improving the overall segmentation accuracy. To demonstrate the effect of each module ablation experiment in this study, we selected the segmentation effects of three different scenes in Area 6 for display, as shown in Figure 6.

Figure 6 shows the visualization of the results of some scenarios in S3DIS. Each column in the figure represents a scenario, and each row represents the corresponding result of each module. The first column on the left shows the meeting room scene. The board on the wall has similar coordinates and spectral information to the wall. The baseline mistakenly categorized the board as a wall and misclassified some walls as columns. The ES module could relatively evenly send all kinds of information into the network, and the segmentation effect was improved to a certain extent. The PYA module improves the discrimination between board and wall by improving the network’s ability to capture global features. The ATT module more completely modeled the spatial relationship between sampling points and neighborhoods, thereby markedly improving the network’s identification ability. The HYB module selected the optimal output model from max pooling and spatial pyramid pooling, preserving global features while considering local features. Therefore, HYB had a better effect than a single PYA module or the baseline in the segmentation of clutter and bookcase. The full-module network ensures that its segmentation effect is close to the labeled data from two aspects: feature learning and information transfer.

The middle column in Figure 6 shows a corridor scene, which contains objects with similar spectral features: door, wall, column, beam, and clutter. The baseline mistakenly divided parts of the ceiling into clutter and wall, and it confused closely located walls and doors. The ES module improved the segmentation effect to a certain extent by transmitting various types of data with similar proportions, and incorrectly categorized substantially fewer items into the wall class. The PYA module was weaker than the HYB module in its ability to learn the geometric features of the sampling point neighborhood, and incorrectly classified some walls as columns, so the HYB module was considerably better than PYA. The ATT module more accurately identified various objects, but misclassified some doors as clutter. The network model that integrated the three modules could more accurately identify the above targets and obtain segmentation results that were very close to the labeled data.

The right side of Figure 6 shows the conference room scene with more clutter. Because the clutter contains various geometric and attribute features, the baseline divided the board into clutter. The ES module classified part of the window as clutter and part of the clutter as bookcase. The PYA module classified the entire area with different properties from the sofa as table; a similar problem occurred with the ATT module. The HYB module misclassified the board as a wall, but the segmentation effect between the door and wall and between the beam and wall was better than those of PYA and ATT, which is consistent with the above analysis of the results. The full network module divided a few parts of the sofa into clutter and table, but the rest of the sofa was classified correctly. The experimental results in Figure 6 further verify that the combination of the ES, ATT, and HYB modules can substantially enhance the model’s ability to describe the local features of the target. The method proposed in this paper is suitable for targets with complex local geometric features or spectral features. Notably, from the perspective of single local feature learning or information transfer, the segmentation accuracy of the network model is limited, and combining the two can effectively improve the network learning ability and obtain more accurate segmentation results.

#### 4.2.2. Six-Fold Crossover Experiment

The aim of the experiments described in this section was to demonstrate the learning ability and generalization of the proposed method for the entire dataset. Taking the spatial coordinates of the scene points and their RGB information as network input features, we divided the room into 1 × 1 m sub-blocks without considering the elevation distribution during training, and we randomly selected 2048 points for each sub-block to generate training data. We selected 12 popular and classic deep learning point cloud semantic segmentation methods. We performed a standard six-fold crossover experiment on the S3DIS dataset. The evaluation indicators of OA and MIoU are shown in Table 6.

Table 6 shows that compared with the baseline, the accuracy of the proposed point cloud deep learning network model based on the balanced sampling and hybrid pooling increased by 9.5% and 11.6% for OA and MIoU, respectively. Compared with other networks (DMF, FPConv, PointWeb, PointCNN, etc.), the OA of the proposed method was the highest and was 0.6% higher than the second best; its MIoU was second best but only 1.1% different from the best. As shown in Table 6, 3DRCNN integrates the pyramid pooling module into the RNN to achieve point cloud semantic segmentation. This method focuses on the learning of local spatial dependencies and lacks the comprehensive learning of global–local features. Therefore, the MIoU of this network was low. DGCNN introduces a dynamic graph convolution module to the CNN infrastructure, which focuses on the local features of various objects and does not grasp the global contours. Therefore, the network failed to obtain a better segmentation effect. NormNet uses a multiscale neighborhood feature learning network based on PointNet for the case of noise disturbance and insufficient samples, realizing the synthesis of global–local features. However, the network lacks salient feature extraction and aggregation abilities, so the segmentation accuracy of the network does not increase due to the overly complex feature information. SPGrap uses superpoints to extract the initial structural features of the point cloud, and sends them to a graph convolutional network for edge feature extraction. This method comprehensively uses context information to improve the segmentation accuracy of the network, but it lacks a suitable pooling module to transmit and express the extracted features. LSANet [35] includes a local spatial awareness layer to capture the local fine-grained features of the sampling center point, thereby improving the segmentation accuracy of the network. However, this method lacks the ability to learn the relationships between neighboring points, so the improvement in segmentation accuracy is limited. Based on CNN, PointCNN applies a learning matrix that considers both spatial coordinate information and local spatial structure. However, this method focuses on improving the network’s ability to learn point cloud features and ignores the problem of missing features caused by imbalanced samples. PointWeb includes an adaptive feature enhancement module and embeds the module into PointNet++’s multiscale feature learning framework to segment point clouds. This method focuses on learning the relationships between the features of each class, so its MIoU performance was better, but part of the OA performance was lost. FPConv [36] applies a 3D object surface convolution operator in a 2D CNN and maps each sampling point and its neighborhood points to a 2D grid for local feature learning. Although 3D mapping to 2D endows the point cloud structural features to facilitate network learning, it destroys the inherent geometric relationship of point cloud 3D data and loses large amounts of key spatial information in the process of dimensional transformation, thus affecting the accuracy of semantic segmentation. DMSF [37] uses an expanded multiscale fusion network to improve MIoU performance by enhancing the receptive field of the network. However, the network also compromises on OA, as does PointWeb. Randla-Net [38] is a large-scale 3D point cloud semantic segmentation network with a random sampling algorithm combined with an attention mechanism. It achieved the best MIoU score of 70% while ensuring sampling density (rich number of point clouds). However, the attention mechanism of the network focuses on learning the local structural features between the center of the sampling point and its neighboring points, ignoring the learning of the mutual structural relationship between the neighboring points. Additionally, the role of the self-attention mechanism in the transfer process of the feature information network is rarely considered, so the OA value of RandLA-Net only ranks fourth.

In summary, simultaneously obtaining the best OA and MIoU is difficult due to the extreme imbalance in points in each category in the S3DIS dataset (as shown in Table 1). When ignoring the minority class samples, the network can obtain a high OA, but MIoU decreases. Conversely, improving the MIoU performance requires the model to pay too much attention to the representative features of the minority class samples, which easily leads to overfitting of the model and limits the overall segmentation accuracy. Therefore, the weight-based balanced sampling module proposed in this paper is appropriate for network feature learning for each type of sample. The self-attention model feature encoding module enhances features from three aspects: local fine-grained feature extraction, contextual information fitting, and spatial relationship learning. Finally, the robust features are sent to the hybrid pooling module for optimal transmission of global–local features. Overall, the proposed algorithm network achieves a satisfactory balance between OA and MIoU.

### 4.3. Network Performance Comparison Based on Different Sampling Parameters

To further verify the ability of the proposed network to learn the characteristics of different density point cloud data, we set the number of sampling points to 2048, 4096, 8192, 16,384, and 32,768, separately, and the corresponding MIoUs are shown in Figure 7. Figure 7 shows that overall sparse sampling corresponds to a lower MIoU, but comparing PointNet++ and the proposed network, we found that when the number of input point clouds was 4096, the overall accuracy of the proposed network substantially improved, up to 9.8%. When the point cloud density increased, the improvement effect was only 8.2%, which shows that the proposed network can effectively improve the ability of a grid model to learn the characteristics of sparse point cloud data.

RandLA-Net is a point sampling method that is most different from the PointNet++ network framework. It combines a random sampling algorithm and an attention mechanism module. It is a lightweight point cloud semantic segmentation network that has attracted much attention in recent years. Its performance largely depends on the density of sampling points. When the density of sampling points is close to that of the network in this study, its segmentation accuracy was lower than that of the proposed network model. When we used 2048 and 4096 sampling points with RandLA-Net, the MIoU was 62.8% and 65.05%, respectively; the proposed network obtained an MIoU of 80% for both. However, with further increases in the number of sampling points, the segmentation performance of RandLA-Net gradually improved, and we obtained the accuracy reported in the literature with 32,768 sampling points. Based on this, we considered that the proposed network model is better in learning the spatial local structure information of sparse sampling points.

### 4.4. Vaihingen Dataset Experiment

To compare the segmentation effects of our proposed method in the training area with those of other methods, we selected nine methods with the best segmentation effects on the IareSPRS official website and recent publications. We calculated the F1 scores and OA on various objects, as shown in Table 7, and the visualization effects of some methods and our method is shown in Figure 8.

Table 7 shows that the OA and MIoU of the proposed method increased by 4.2% and 3.9%, respectively, compared with those of the baseline. Compared with other methods, it obtained the best OA score and the fifth best MIoU. Our method also achieved the best results in two categories, roof and tree, and the worst performance for hedges. The DPE network focuses on feature learning for the impervious surface category. WhuY4 [39] describes the characteristics of local minority class samples through multiscale point cloud feature maps, and it achieved the best results for the hedge class but was less accurate for the other classes. NANJ2 [40] combines decision trees with multiscale point cloud feature maps and achieved the best results in the shrub category, but was less accurate in other categories, similar to WhuY4. The D-FCN [41], DANCE-Net [42], and GACNN [43] models use their core modules to effectively learn minority class structural features, thereby obtaining a higher average F1 score but a lower OA. Both GraNet and GANet models use an attention mechanism to score and assign importance to features, which helps the network to identify features and suppress interference, so the two methods had higher overall segmentation accuracy and average F1 score.

## 5. Conclusions

In 3D point clouds, semantic segmentation is a challenging fine-grained task for network models to correctly classify all the input points. Currently, the deep learning networks for direct point cloud semantic segmentation generally sample the original point cloud to meet the needs for large-scale data processing and network model input. However, the uneven number of point clouds of various objects and the uneven sampling density of sub-blocks affect the adjacency relationship of the original target points, thereby impacting the ability of the network model to capture fine-grained local features. Therefore, with the loss function as the constraint, the sampling method of adjusting the sub-block sampling weight can effectively improve the possibility of minority class samples being captured. Notably, the network we designed in this study effectively enhances the ability of the network to learn the saliency structures (features) from the sampling point neighborhood by more completely and meticulously observing the spatial relationship in the neighborhood of the sampling point and ensuring that important information is transmitted as much as possible. However, we only verified and analyzed the algorithm on the S3DIS indoor dataset and on an urban dataset obtained by airborne aerial photography; we did not verify or analyze the method on other types of outdoor datasets, such as mountain forest, atmosphere, and close-up scenery. Improving the generalization and robustness of this algorithm in other outdoor datasets is the direction of future research.

## Figures and Tables

**Figure 1 sensors-23-00981-f001:**
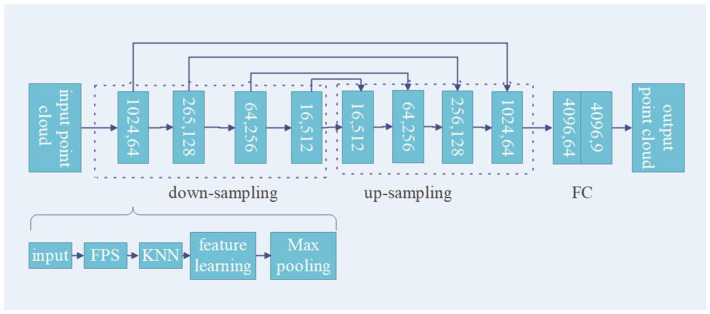
Structure of PointNet++.

**Figure 2 sensors-23-00981-f002:**
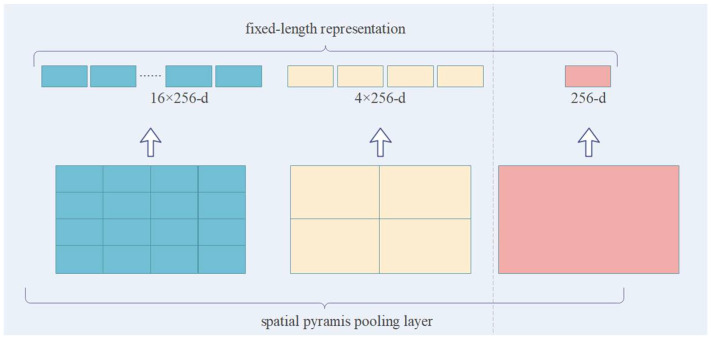
Spatial pyramid pooling framework.

**Figure 3 sensors-23-00981-f003:**
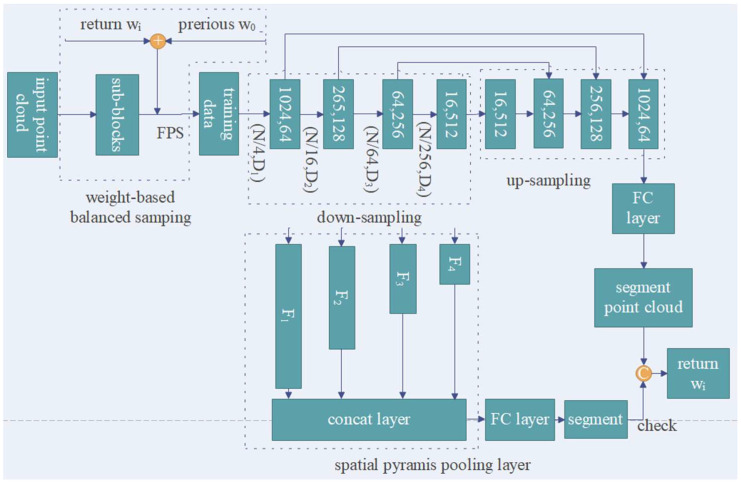
Overall structure of network.

**Figure 4 sensors-23-00981-f004:**
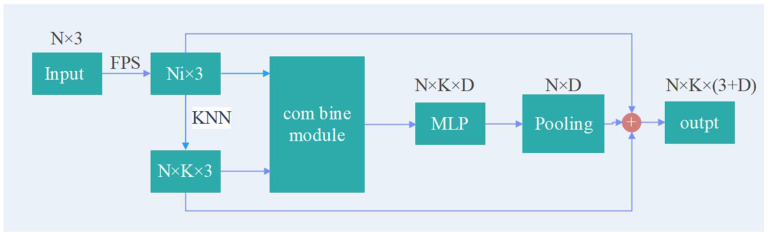
SC feature learning module for the downsampling layer.

**Figure 5 sensors-23-00981-f005:**
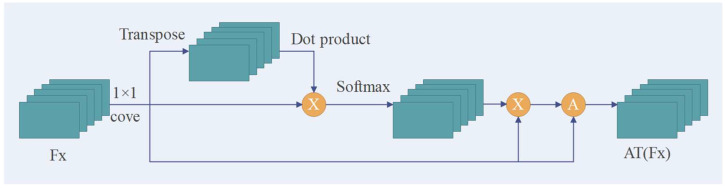
Attention model feature enhancement module.

**Figure 6 sensors-23-00981-f006:**
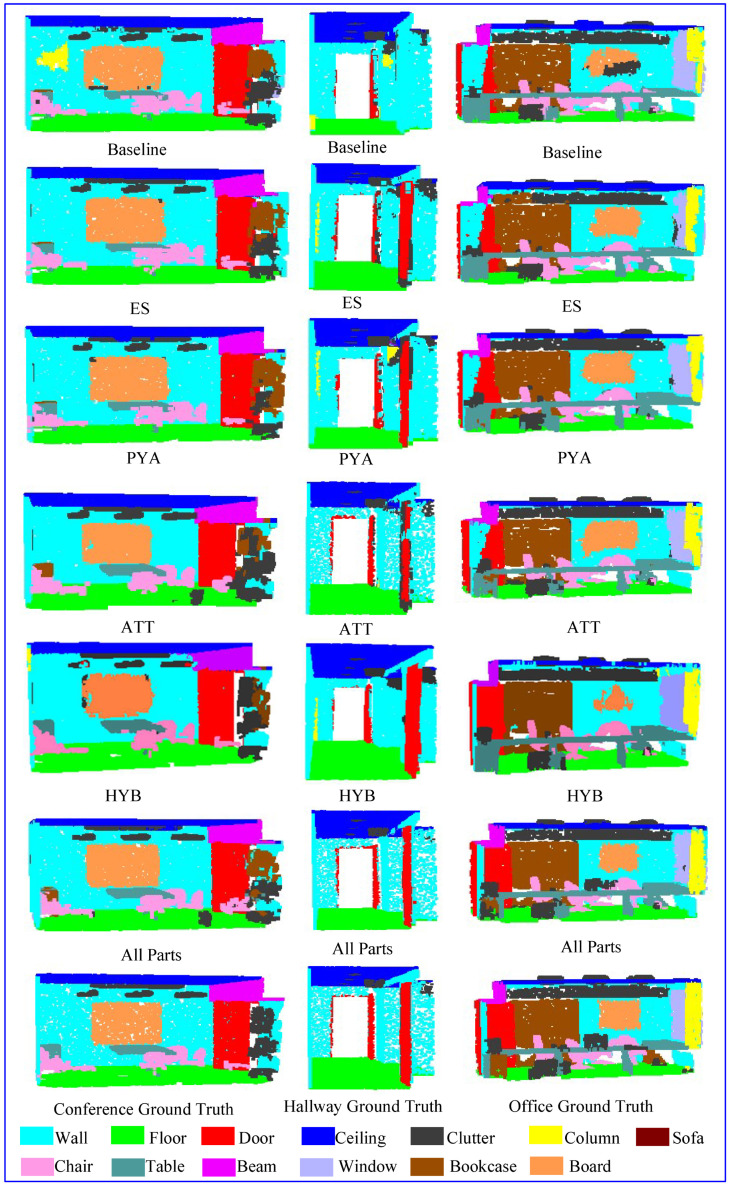
Comparison of visualization results of partial scene segmentation.

**Figure 7 sensors-23-00981-f007:**
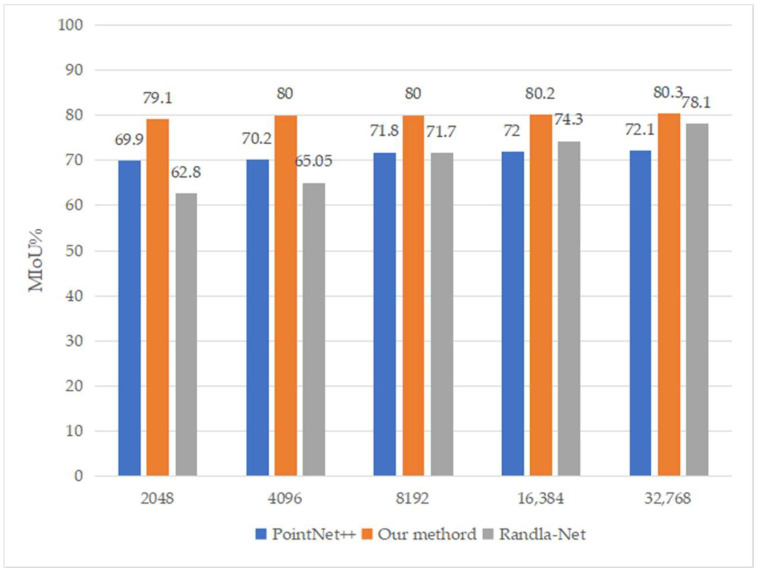
Overall segmentation accuracy based on different sampling parameters in Area 6.

**Figure 8 sensors-23-00981-f008:**
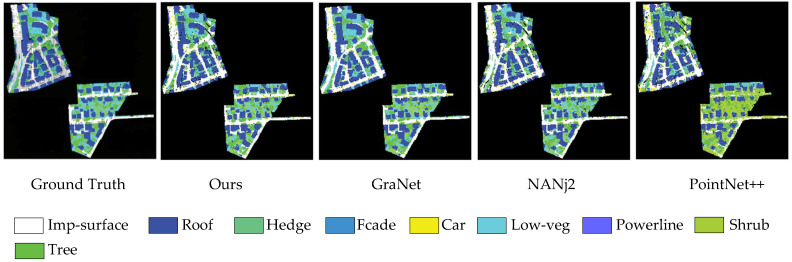
Visualization results produced our method and other models.

**Table 1 sensors-23-00981-t001:** Areas 1–5 dataset details (**%**).

Class	Proportion	Class	Proportion
ceiling	21.6	table	2.7
floor	19.4	chair	3.6
wall	26.0	sofa	0.4
beam	1.2	Bookcase	5.5
column	1.5	board	1.0
window	2.0	clutter	9.8
door	5.3		

**Table 2 sensors-23-00981-t002:** Details of training data set.

Class	Power Line	Car	Façade	Hedge	Impervious Surface	Low Vegetation	Roof	Shrub	Tree
Training	546	4614	27,250	12,070	193,723	180,850	152,045	47,605	135,173
Proportion	0.072%	0.612%	3.615%	1.601%	25.697%	23.989%	20.168%	6.315%	17.931%

**Table 3 sensors-23-00981-t003:** Module details.

Name	Module
PointNet++	Baseline
+ES	Weight-based balanced sampling
+ATT	SC Feature encoding
+PYA	Spatial pyramid pooling
+HYB	Hybrid pooling
ALL	Our method

**Table 4 sensors-23-00981-t004:** Statistical results of evaluation indicators of each module (%). The best results are presented in bold.

Module	MIoU	OA	Ceiling	Floor	Wall	Beam	Column	Window	Door	Table	Chair	Sofa	Bookcase	Board	Clutter
Baseline	70.2	87.7	93.0	97.3	74.8	68.7	43.2	77.8	78.9	72.4	76.8	41.9	58.7	66.2	63.2
ES	70.7	88.1	**94.4**	96.5	76.4	52.9	47.1	74.3	70.2	74.7	77.8	56.1	62.4	68.6	67.8
PYA	72.1	89.2	93.8	**97.5**	79.1	66.2	48.5	74.3	78.3	74.9	77.3	44.4	65.0	72.1	66.0
ATT	73.3	89.6	93.3	97.1	80.1	66.0	50.9	73.2	83.5	73.7	80.2	49.3	65.7	73.9	65.6
HYB	75.8	90.9	94.1	97.4	**82.8**	69.2	60.4	80.5	**84.4**	77.0	78.5	46.6	69.2	76.8	69.2
ALL	**80.0**	**93.4**	**94.4**	**97.5**	82.2	**74.6**	**63.4**	**83.9**	83.3	**80.0**	**87.7**	**64.9**	**75.6**	**79.0**	**73.3**

**Table 5 sensors-23-00981-t005:** Effects of each component on the segmentation performance.

**ES**	**PYA**	**ATT**	**HYB**	**OA**	**MIoU**
	√	√	√	94.0	77.1
√		√	√	89.2	75.4
√	√		√	88.9	73.3
√	√	√		90.0	74.9
√	√	√	√	93.4	80.0

**Table 6 sensors-23-00981-t006:** Semantic segmentation accuracy on S3DIS dataset.

Method	OA	MIoU
PointNet [1]	78.5	47.6
3DRCNN [9]	85.7	53.4
PointNet++ [5]	81.0	54.5
DGCNN [15]	84.1	56.1
NormNet [11]	84.5	57.1
SPGrap [16]	85.5	62.1
LSANet [35]	86.8	62.2
PointCNN [10]	88.1	65.4
PointWeb [12]	87.3	66.7
FPConv [36]	89.9	66.7
DMSF [37]	87.9	67.2
Randla-Net [38]	88.0	70.0
Ours	90.5	66.1

**Table 7 sensors-23-00981-t007:** Comparison of F1 score and OA of different methods (%).

Model	Power Line	Car	Facade	Hedge	Impervious Surface	Low Vegetation	Roof	Shrub	Tree	OA	Average F1
PointNet++ [5]	57.9	66.1	54.3	31.5	90.6	79.6	91.6	41.6	77.0	81.2	65.6
DPE [14]	68.1	75.2	44.2	19.5	99.3	86.5	91.1	39.4	72.6	83.2	66.2
WhuY4 [39]	42.5	74.7	53.1	53.7	91.4	82.7	94.3	47.9	82.8	84.9	69.2
NANJ2 [40]	62.0	66.7	42.6	40.7	91.2	88.8	93.6	55.9	82.6	85.2	69.3
D-FCN [41]	70.4	78.1	60.5	37.0	91.4	80.2	93.0	46.0	79.4	82.2	70.7
DANCE-Net [42]	68.4	77.2	60.2	38.6	92.8	81.6	93.9	47.2	81.4	83.9	71.2
GACNN [43]	76.0	77.7	58.9	37.8	93.0	81.8	93.1	46.7	78.9	83.2	71.5
GANet [17]	75.4	77.8	61.5	44.2	91.6	82.0	94.4	49.6	82.6	84.5	73.2
GraNet [18]	67.7	80.9	62.0	51.1	91.7	82.7	94.5	49.9	82.0	84.5	73.6
Our method	46.5	77.8	57.9	37.9	92.9	82.3	94.8	48.6	86.3	85.4	69.5

## Data Availability

The S3DIS data set can be found in “http://buildingparser.stanford.edu/dataset.html”, or it can be downloaded from “https://drive.google.com/drive/folders/0BweDykwS9vIoUG5nNGRjQmFLTGM?resourcekey=0-dHhRVxB0LDUcUVtASUIgTQ”. The ISPRS Vaihingen 3D semantic segmentation competition dataset can be found in “https://www.isprs.org/education/benchmarks/UrbanSemLab/Default.aspx”.

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
