# Peer review of "Point Cloud Deep Learning Network Based on Balanced Sampling and Hybrid Pooling"

_sensors, 2023, doi:10.3390/s23020981_

Round 1
Reviewer 1 Report
This paper designs a weighted sampling method based on Farthest Point Sampling (FPS). The approach includes three modules, a weight-based balanced sampling, a self-conv (SC) module, and a hybrid pooling method. The proposed network achieves 9.5% and 11.6% improvement in Overall point-wise Accuracy(OA) and Mean of class-wise Intersection Over Union (MIoU) compared with the baseline.
.
Major revision:
1. The structure of this manuscript is not clear and some important parts are missing. For example, the overview pipeline figure and description are missing, Without the overview pipeline figure with description, it is hard to figure out what is the major contribution of this work.
2. what is the relationship between these four modules. that is, ES, ATT, PYA, HYB. Does they all show positive and same effects on the segmentation results. More experiments on ablation experiments are required.
3. Why use Pointnet++ as the based line despite there are more advanced and recent models. The details of the Pointnet++ have to be illustrated.
4. Visualization Results of the proposed method and compared models need to be added.
5. How about the segmentation performance on other public data, such as the scannet that utilized originally in the Pointnet++ paper. More dataset are required to further verify the proposed models.
some minor revision:
1. The subsection of the 2 related work is “subsection”? Strongly suggest the author to read through the whole manuscript several times.
2. Some grammar mistakes, for example, Line337, a space is missing. The author should pay more attention to the typo errors.
3. References are very old and not enough. There are only few references in recent 2 years. And the latest SOTA works in point cloud detection are missing. Please rewrite the related work section.
Author Response
Dear Reviewer:
Thank for your comments and suggestions!
We have revised the paper according to your suggestion!More details are as follow:
1.We redraw the overall structure of the network,it can be seen in page 8 which named Figure 3.
2.We add a new ablation experiment in page 12, it can be seen in page 12 which named Table 5.
3.We add structure of PointNet++ in page 4.
4.We have added visualization results in page 18
5.We also add a new segmentation performance on Vaihingen dataset, it can be seen in page 17.
6.We have added 14 new references
7.We has undergone English language editing by MDPI.

Reviewer 2 Report
Within the scope of the study, a new deep learning-based point cloud semantic segmentation approach is proposed. Although the study addresses an interesting and popular area, the manuscript needs major revision. The language of the whole manuscript should be revised from top to down. However, I have some minor comments related to the manuscript. Here are my suggestions:
Comments related to the manuscript are as follows;
The literature review of the study should be expanded by adding recently published studies.
A literature review is presented in both the Introduction and the Related Work sections. These studies can be explained in one section. This might be the section of Related Works section.
The title of Section 3 should be changed to "Material and Methods". First, the data set used should be explained. Then the proposed method should be presented. Authors should re-manage the manuscript considering the consistency.
Equations should be written following the format of the journal.
The legend should be added to Figure 5. Class colors must be specified. Additionally, the first letters of the words in the figure title should be capitalized.
When discussing, methods that are referenced once do not need to be referenced again. For example, Based on CNN, PointCNN [7] proposes… (in line 490)
In the Conclusions section, suggestions for future studies should be added to improve the technical quality of the manuscript.
Can the study be extended to outdoor datasets? Has a study been done on this? The authors should provide the necessary information about the choice of an indoor dataset to work.
The study can be applied to different data sets. Thus, it can be shown that the method can be generalized. It should be investigated whether the results will be similar in other data sets.
Author Response
Dear Reviewer:
Thank for your comments and suggestions! We have revised the paper according to your suggestions!More details are as follows:
1.We have added recently published studies as references in page 20.
2. We have rewritten introduction and relate work.
3. We renamed the third section as "Material and Methods", and we also added dataset introduction in section 3.
4. All Equations have be written following the format of the journal.
5. Figue 5 has been redrawn and named Figure 6.
6. The problem of literature being cited again has been solved.
7. Suggestions for future studies has been added in the Conclusions section.
8. We have added Vaihingen dataset(outdoor dataset) experiment.
9. We has undergone English language editing by MDPI.

Round 2
Reviewer 1 Report
I have no more comments
Reviewer 2 Report
Due to the track change option, there are some grammatical errors but I think they are going to be corrected after acceptance of all changes. The authors realized almost all my comments related to the previous version of the manuscript. Thus, it might be published as it is.